# Variability and Plasticity in Cuticular Transpiration and Leaf Permeability Allow Differentiation of *Eucalyptus* Clones at an Early Age

**André Carignato [1], Javier Vázquez-Piqué [1] , Raúl Tapias [1], Federico Ruiz [2] and Manuel Fernández [1,*]**

[1] Department of Agroforestry Sciences, School of Engineering, University of Huelva, 21071 Huelva, Spain; andrecarignatoflorestal@hotmail.com (A.C.); jpique@dcaf.uhu.es (J.V.-P.); rtapias@dcaf.uhu.es (R.T.)

[2] ENCE, Energía y Celulosa, S.A. (ENCE)-Ctra A-5000, km. 7, 5. Apdo. 223, 21007 Huelva, Spain; fruiz@ence.es

[*] Correspondence: manuel.fernandez@dcaf.uhu.es

**Abstract:** *Background and Objectives.* Water stress is a major constraining factor of *Eucalyptus* plantations' growth. Within a genetic improvement program, the selection of genotypes that improve drought resistance would help to improve productivity and to expand plantations. Leaf characteristics, among others, are important factors to consider when evaluating drought resistance evaluation, as well as the clone's ability to modify leaf properties (e.g., stomatal density ($d$) and size, relative water content at the time of stomatal closure ($RWC_c$), cuticular transpiration ($E_c$), specific leaf area ($SLA$)) according to growing conditions. Therefore, this study aimed at analyzing these properties in nursery plants of nine high-productivity *Eucalyptus* clones. *Material and Methods*: Five *Eucalyptus globulus* Labill. clones and four hybrids clones (*Eucalyptus urophylla* S.T. Blake × *Eucalyptus grandis* W. Hill ex Maiden, 12€; *Eucalyptus urograndis* × *E. globulus*, HE; *Eucalyptus dunnii* Maiden–*E. grandis* × *E. globulus*, HG; *Eucalyptus saligna* Sm. × *Eucalyptus maidenii* F. Muell., HI) were studied. Several parameters relating to the aforementioned leaf traits were evaluated for 2.5 years. *Results:* Significant differences in stomatal $d$ and size, $RWC_c$, $E_c$, and $SLA$ among clones ($p < 0.001$) and according to the dates ($p < 0.001$) were obtained. Each clone varied seasonally the characteristics of its new developing leaves to acclimatize to the growth conditions. The pore opening surface potential (i.e., the stomatal $d$ × size) did not affect transpiration rates with full open stomata, so the water transpired under these conditions might depend on other leaf factors. The clones HE, HG, and 12€ were the ones that differed the most from the drought resistant *E. globulus* control clone (C14). Those three clones showed lower leaf epidermis impermeability (HE, HG, 12€), higher $SLA$ (12€, HG), and lower stomatal control under moderate water stress (HE, HG) not being, therefore, good candidates to be selected for drought resistance, at least for these measured traits. *Conclusions*: These parameters can be incorporated into genetic selection and breeding programs, especially $E_c$, $SLA$, $RWC_c$, and stomatal control under moderate water stress.

**Keywords:** early selection; stomatal characteristics; water stress; water relations; specific leaf area; *Eucalyptus* clones

## 1. Introduction

Leaf morphology (e.g., specific leaf area, $SLA$), stomatal characteristics (e.g., stomatal size and density ($d$)) and stomatal opening are closely linked to physiological activity and transpiration control, in turn related to growth and survival. Responding to the available resources, plants can adjust these

characteristics and acclimatize to changing environmental conditions [1,2], and this acclimatization process is genetically influenced [3,4]. Nevertheless, the relationships between stomatal size, *d* and stomatal conductance (gs) should be treated with caution, because the speed of variation of gs is not necessarily related to *d* or stomatal size in different species, or individuals within a species [5].

The regulation of stomatal opening is multigenic, resulting in a multiple control mechanism (water status, illuminance, vapor-pressure deficit (VPD), photosynthetic activity, $CO_2$ concentration, etc.). However, how simultaneous stomatal signals interact and influence stomatal behavior is relatively unexplored [6]. The tightness of stomatal closure also constitutes an important component of stomatal control, particularly during a drought when it is necessary to restrict transpiration water-loss [7]. When exposed to drought, stomata can become increasingly sensitive to $CO_2$ concentration and leaf abscisic acid concentration (ABA) in comparison to photosynthetically active radiation (PAR), leaf to air VPD, and leaf water potential [6]. Consequently, water loss through the stomata is based on guard cells opening variations [2], which are in turn produced by fluxes of potassium ions ($K^+$) in or out of the guard cell [8]. Such mechanisms could include physiological de-/activation of ion transport in the stomatal guard cells, or a genetic control of the expression of ion transport channels [9]. When an area of leaf is placed under conditions of high leaf to air VPD stomata usually close to prevent desiccation, while low leaf to air VPD is conducive to higher rates of gs as the potential of excessive water-loss is diminished [10,11]. Species that have more effective stomatal control are therefore expected to withstand water deficit situations more successfully. However, not all of them have equally effective stomatal control, whether in terms of number of stomata during leaf development or of their regulation of stomatal opening [12]. However, stomatal behavior is not universal, with some species altering the number of stomata on newly developed leaves in response to ($CO_2$) rather than utilizing physiological regulation of stomatal aperture [13]. Stomatal density and size are usually highly sensitive to environmental abiotic stress such as drought because of stomatal resistance to transpiration [14,15]. Stomatal density (*d*) is often negatively related to stomatal size [16]. A signaling mechanism from the mature leaves to the developing leaves seems to exist, leading to the optimization of *d* and of stomatal size to face the changes to come in future environmental conditions [17].

The *SLA*, for its part, indicates how the leaf biomass is distributed, seeking a balance between carbon gain and water loss, since at equal mass, a broader and thinner leaf blade favors not only photosynthesis but also water loss by transpiration. This property plays an important role in allowing plants to adapt to environmental conditions and this plasticity is often understood as a way of optimizing light absorption as well as water use efficiency (WUE) [3,18]. Thus, as they develop their leaves, plants can resort to certain morphological alterations such as palisade parenchyma thickness [19] or epidermis and cuticle water-tightness [20], the latter being essential to control water losses when stomata are closed, especially during drought periods, by means of so-called cuticular transpiration ($E_c$).

In areas with marked seasons, especially in regions that have dry seasons, such as the Mediterranean, evaporative demands vary considerably throughout the year and plants must constantly acclimatize: it is thus interesting in these cases to understand how plants transpire over a complete annual cycle [21]. The *Eucalyptus* genus stands out as one of the most widely planted exotic genera in tropical and Mediterranean climate regions and, together with *Pinus*, represents 98% of the world's forestry production [22,23]. Within the genetic improvement programmes of this genus, it is possible to associate desirable characteristics of different species by synthesizing interspecific hybrids [24] and, together with the cloning technique, to generate homogeneous plantations that are highly productive and resistant to pests and diseases [25,26]. Therefore, it is necessary to incorporate simple tools in the evaluation methods of a genetic improvement program in order to evaluate the genotypes and that these tools be applicable on a large scale (i.e., that they measure easily and quickly). For instance, it is essential today to select taxa of *Eucalyptus* spp. that are resistant to water deficit, mainly in regions subject to irregular and scarce annual rainfall regimes [27], and at an early age in order to shorten any improvement program [22]. Among the most widely planted species of eucalypts, it is known that *Eucalyptus camaldulensis* Dehnh. is a drought-tolerant species [28], *Eucalyptus globulus* offers certain

genotypes that potentially tolerate environments with low water availability [29], and the hybrid *Eucalyptus* × *urograndis* is usually sensitive to water deficit [30,31].

WUE relates photosynthetic rate or plant growth to water consumption and has been shown to be a useful physiological parameter to assess plant drought adaptation [22,32,33], and to differentiate the behavior of different taxa [28]. However, WUE is not always a constant trait of a given taxon: it varies according to a specific combination of conditions of the site, weather, and tree age [34]. While the physiological characteristics may vary within a very short interval, the morphological characteristics do not. They maintain the same structure while the organ is functional despite possible environmental changes. Hence the importance of developing organs with an appropriate structure to withstand coming environmental conditions and of studying the effect of some anatomical leaf characteristics on water loss due to plant transpiration, with either fully open or totally closed stomata. For example, anatomical structures of leaves such as palisade parenchyma and stomatal density, and leaf morphology such as leaf thickness and specific leaf weight, regulate the physiological functions (i.e., photosynthesis and transpiration), which vary in different cultivars or clones [35]. Moreover, during drought conditions, gs is directly affected, and the consequent stomatal closure is a way of reducing the water loss due to leaf transpiration and the susceptibility of xylem vessels to cavitation (i.e., embolism or dysfunction) that results in a reduction in hydraulic conductance [36,37]. Therefore, since drought-resistance is a multiple control mechanism, it is the conjunction of several factors, not just one factor, that represents the true degree of each taxon's water consumption and drought-resistance [38,39]. In addition, because annual plant growth and water consumption depend not only the dry season but on the whole year, we hypothesized that differences would exist among genotypes regarding stomatal characteristics and leaf structure, which would vary throughout the year depending on environmental conditions and would be detectable at an early age. The present study deals with nursery plants of nine high productivity *Eucalyptus* clones belonging to a breeding program and that could be used in commercial plantations. It focused on comparing these clones and the seasonal development of (1) leaf stomatal size and density (*d*); (2) cuticular transpiration ($E_c$); (3) and specific leaf area (*SLA*). The objective was to detect traits that could be incorporated into the improvement programs of this plant genus.

## 2. Materials and Methods

### 2.1. Plant Material and Growing Conditions

The starting plant material consisted of 10-month-old plants of five clones belonging to *E. globulus* (reference codes: C14, 225, 227, 358, 437) and four hybrid clones (12€, *Eucalyptus urophylla* S.T. Blake × *Eucalyptus grandis* W. Hill ex Maiden; HE, *E. urograndis* × *E. globulus*; HG, *Eucalyptus dunnii* Maiden–*E. grandis* × *E. globulus*; HI, *Eucalyptus saligna* Sm. × *Eucalyptus maideni* F. Muell.), obtained by rooting cuttings in a commercial nursery, in 150 cm$^3$ containers, provided by the plant selection and breeding program of the ENCE, energía y celulosa, Inc. (Madrid, Spain) The first clone, C14, belonged to the first generation ($F_0$). It offers high productivity and plasticity and is widely used by the company in commercial plantations, even in areas subject to dry summer seasons. The others corresponded to clones of later generations of improvement and field trials have shown that they can increase productivity by up to 25% with respect to C14 under favorable growing conditions. However, no significant differences were obtained between clones for plant growth in the three assays performed in this study under nursery conditions (data not shown). Generally, within the *Eucalyptus* genus, *E. grandis*, *E. dunnii*, and *E. saligna* are described as low drought-resistant species, while *E. globulus* and *E. urophylla* are moderately resistant [22,34,40,41]. According to field trials with 4–10 year-old plants, the ENCE company classifies seven clones for their drought resistance following this approximate ranking: C14, 437 ≥ 227, 358, 225 ≥ HE, 12€. Further information, however, is not available.

For three consecutive years, they were transplanted in December into 10 L containers allowing for a full vegetative period. Each year, the experimental design consisted of four randomly distributed plants per clone (36 plants per year, 108 plants in total). The substrate consisted of a mixture of peat,

coconut fiber and perlite (2:2:1 by volume), that was well watered to field capacity and fertilized using Ferticote 16-7-8 + 2 MgO + Micros (Burés profesional S.A., Girona, Spain) applying a dose of 1.5 kg m$^{-3}$. Four additional plants per clone and year were grown under the same conditions in order to replace any of the tested plants in case of need. No plants, however, died during the experimental period. The plants were placed outdoors and fully exposed to sunlight in an experimental plot at the University of Huelva (37°12′03″ N, 6°54′53″ W, 5 m a.s.l.).

*2.2. Stomatal Characteristics*

Across the four seasons of the year, at 10 different dates, and for 2.5 years in a row, 3–4 fully developed leaves were collected per clone and season (i.e., the leaves developed during spring, summer, autumn, or winter were collected at every measurement date), from the 3rd to the 5th whorl of the main stem (34 leaves per clone, 306 leaves in total during the period of study). In the case of *E. globulus* clones, which present foliar dimorphism between mature and juvenile leaves, the harvested leaves were always juvenile due to the plants' size and age. To select the sample leaves, we considered hardness to the touch, the growth stoppage and the constant value of the *SLA.* For this, previous tests were carried out and the additional plants were used to verify these characteristics. Each year, on the dates of the first measurement, the plants averaged 6 mm in stem diameter and 60 cm in height, while on date of the last measurements, after a vegetative period, they averaged 17 mm and 150 cm, respectively. The leaf samples were considered to have developed during the 90-day period prior to each measurement date. This foliar development is highly influenced by environmental conditions (light radiation, humidity and VPD, temperature, etc.), which in turn affect the physiological state of the plant, showing phenotypic plasticity [1,4,18–20,42]. The leaf´s morphology and internal structure are formed during its growth and development and, once fully developed, its anatomy does not vary substantially during the rest of its life. Therefore, growth conditions during the leaf's development are more interesting to consider than the conditions of the rest of the year or of subsequent months. Table 1 shows the values of the relevant climatic variables in the area during the study.

**Table 1.** Temperature, relative humidity, and solar radiation in the nursery 90 days prior to each measurement date: the measurements of February, May, and November were made at the beginning of each month, and the July measurements were made at the end of the month.

| Variables | Measurement Dates | | | | | | | | | |
|---|---|---|---|---|---|---|---|---|---|---|
| | Feb 2015 | May 2015 | July 2015 | Nov 2015 | Feb 2016 | May 2016 | July 2016 | Nov 2016 | Feb 2017 | May 2017 |
| $T_{90}$ [a] (°C) | 16.3 | 19.9 | 30.0 | 24.9 | 19.8 | 19.3 | 28.6 | 27.3 | 17.9 | 20.9 |
| $t_{90}$ [b] (°C) | 3.3 | 6.9 | 15.1 | 14.0 | 8.5 | 7.5 | 14.4 | 13.8 | 5.7 | 8.0 |
| $RH_{90}$ [a] (%) | 102.1 | 100.1 | 89.6 | 95.8 | 95.2 | 93.6 | 86.7 | 86.9 | 89.8 | 91.6 |
| $rh_{90}$ [b] (%) | 59.0 | 52.3 | 33.1 | 51.0 | 52.7 | 45.9 | 37.0 | 42.8 | 50.5 | 46.3 |
| $R_{90}$ [c] (MJ m$^{-2}$ d$^{-1}$) | 9.3 | 16.2 | 26.3 | 16.0 | 9.1 | 16.8 | 26.2 | 17.1 | 9.5 | 16.2 |

[a] $T_{90}/RH_{90}$: average maximum daily temperature/relative humidity over the 90-day period. [b] $t_{90}/rh_{90}$: average minimum daily temperature/relative humidity over the 90-day period. [c] $R_{90}$: cumulative daily solar radiation over the 90-day period.

Leaf prints of these leaves were collected using nail varnish, and the stomata could be observed. Having detected the absence or extreme scarcity of the stomata on the leaves' adaxial side, prints were made of three zones (basal, B; central, C; and apical, A) on the abaxial side (Figure 1a). The image recorded in the nail varnish prints was mounted on a slide that allowed viewing under optical microscope (Leica DM/LS, Leica Microsystems) using an image capture software (Leica LAS EZ, Leica Microsystems). Two images magnified 100 times were randomly captured of each zone (six images per leaf) to determine *d* (number of stomata per mm$^2$). Subsequently, the number of stomata present in five randomly selected squares measuring 250 × 250 μm (0.0625 mm$^2$) on each image were counted. Since each image was 1.05 mm$^2$ (1.200 × 0.875 mm), the area measured in the five grids accounted for

29.8% of the image taken. A total of 30 grids per leaf (10 grids per zone) were measured. Five images magnified 400 times were also randomly taken to define the width and length both of the stomatal cells (*SW* and *SL*, respectively) and of the epidermis opening where the ostiole was located (*OW*, *OL*) (Figures 1b and S1). To determine stomatal size, 93 randomly chosen stomata on each leaf (31 stomata per zone) were measured.

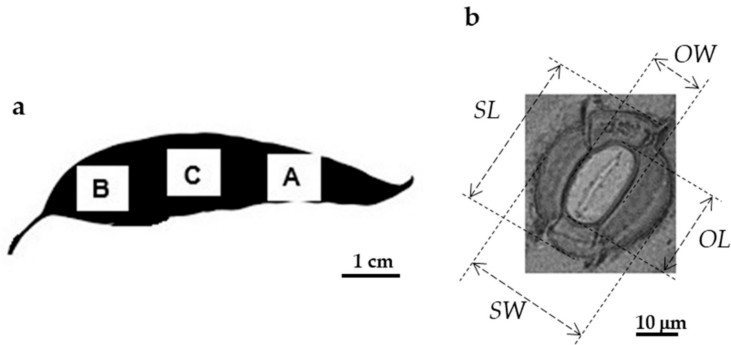

**Figure 1.** (**a**) Leaf zones (B; C; A) of which leaf prints were made to observe the stomata, and (**b**) the width and length of the stomatal cell (*SW*, *SL*) and the epidermis opening where the ostiole is located (*OW*, *OL*).

## 2.3. Cuticular Transpiration

On the same dates the stomata were characterized, three more leaves with similar characteristics to those used for the stomata were taken per clone in order to measure $E_c$ under laboratory conditions. The leaves were sampled 1–2 h after dawn, presenting an apparently good water status. Just after cutting, they were placed in sealed plastic bags and taken to the laboratory chilled in a portable cooler. Once in the laboratory (less than 20 min after they were sampled), and following a standard methodology [43], the leaves were hydrated until saturation (in the dark, at 4 °C for 16 h). The next day, before starting the measurements, the leaves were left in the laboratory in the dark until they reached room temperature. They were then weighed in succession using precision scales (±0.1 mg) at short time intervals (every 5 min during the first hour, every 10 min during the following 2 h and every 20 min thereafter, until weight loss was constant over time). During the measurements, the leaves were exposed to light (430 µmol m$^{-2}$ s$^{-1}$ of PAR, using LED lamps), with their abaxial facing downwards on a grid, to allow free circulation of air on both sides, in an environment conditioned at 20–23 °C and 45%–60% relative humidity. $E_c$ was measured under laboratory conditions, maintaining homogeneous environmental conditions for all leaves and at all measurement dates, so that the data could be compared.

Once the leaf area (*LA*) was measured and using the previously collected data (fresh weight, *FW*; and time), it was possible to determine the time elapsed until stomata closure ($t_c$), estimated by the cut-off point between the curve generated by all value pairs (fresh weight-time) and the regression line generated by the points marking a leaf's constant weight drop (Figure S2); $E_c$ as the slope of the regression line, since the stomata are supposed to be closed when the weight drop is constant; and the relative water content and the moisture content at the time of stomatal closure ($RWC_c$ and $M_c$, respectively). Initially, transpiration occurred through the stomata (stomatal transpiration, $E_s$) and leaf epidermis ($E_c$), but after stomatal closure, only the $E_c$ remained. The leaves were then heated in an oven at 70 °C until they reached a constant weight to determine their dry weight (*DW*). With all these data, it was possible to determine the $E_c$ based on *DW* and *LA*, as well as *SLA* ($SLA = LA/DW$, m$^2$ kg$^{-1}$). Additionally, to know the relationship between gas exchange and leaf water potential ($\Psi$), a test was carried out during the summer of the last year in which the plants were subjected to progressive water stress. These data are shown as supplementary material (Figures S3–S6; Table S1) because they are not the main objective of this study and were measured only at one moment in the year.

## 2.4. Data Analysis

Regarding *d* on the abaxial leaf surface, the data were analyzed following a Generalized Mixed Model with Poisson distribution and a logarithmic link function. Regarding stomatal size (*SW*, *SL*), the data were analyzed by means of a Mixed Linear Model with Gaussian distribution and identity link function. We took into account the fixed Clone, Date, and Clone × Date interaction effects, and the plant nested within the clone as a random effect. The Akaike information criterion (AIC) was used to choose the selected models [44]. The differences between the groups of the different factors were analyzed by conducting a Scheffé test. Regarding $E_c$ and associated parameters, as well as *SLA*, the data were analyzed by means of a General Linear two-factor Model (clone, date), which were considered fixed, and the differences between the groups of the different factors were analyzed using Dunnett's T3 test. The fixed Clone, Date, and Clone × Date interaction effects were taken into account. The statistical package SAS® 9.2 was used. The differences were deemed significant obtaining a significance level of $p \leq 0.05$.

## 3. Results

### 3.1. Stomatal Characteristics

Significant differences relating to *d*, stomatal size (*SL*, *SW*) and the *SW/SL* ratio among clones ($p < 0.001$) and among dates ($p < 0.001$) were detected (Tables 2 and 3). The interaction between the two factors was also significant ($p < 0.001$), indicating a different seasonal development pattern among clones (Figure 2). The results obtained for *SW*, *OW*, and *OL* are not presented in more detail due to the high and significant correlations obtained: *SL* vs. *SW* ($r = 0.746$), *SL* vs. *OL* ($r = 0.795$), *OL* vs. *OW* ($r = 0.683$), 14,043 being the sample size and $p < 0.001$ for all of them. These three parameters showed seasonal development and differentiation among clones and dates similar to *SL*. The mean values (±SE) obtained for the series of clones and measurement dates were *SW* = 16.6 ± 0.3 μm; *OL* = 14.4 ± 0.2 μm; *OW* = 10.1 ± 0.2 μm. The *SW/SL* ratio indicated that the stomata tended to be elliptical, and only the three clones with the lowest average value (HE, HG, and HI) were significantly differentiated from the clone with the highest value (437) (Table 2).

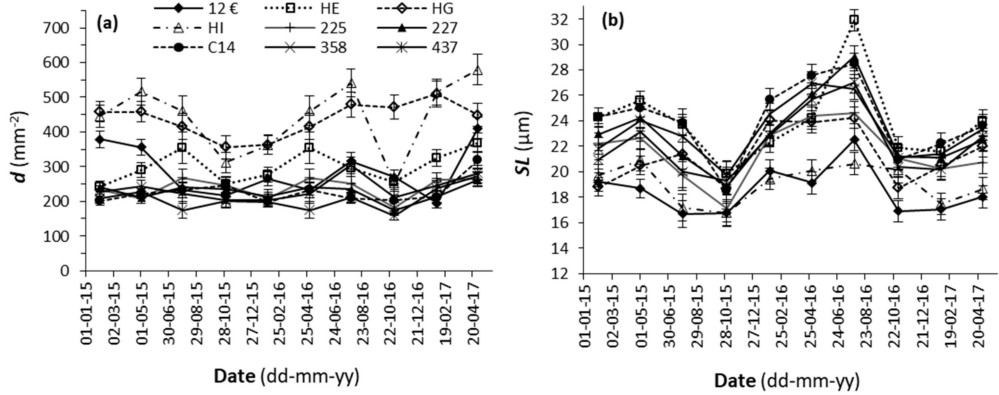

**Figure 2.** Seasonal evolution (average ±SE) of (**a**) stomatal density (*d*), and (**b**) stomatal length (*SL*) of the nine *Eucalyptus* clones studied on the 10 measurement dates.

Taking the maximum possible stomatal opening into account—that is, the share of *LA* that all open ostioles would entirely occupy if they covered the full window left free by the epidermis, calculated via the expression $d \times (\pi \times OW \times OL)/4$—the *E. globulus* clones as well as the 12€ hybrid were in the lowest range (from 2.3% for 12€ to 3.2% for C14), while the other three hybrids (HI, HE, and HG) were within a range of 3.7% (HI) to 4.4% (HG).

**Table 2.** Average values (±SE) of stomatal density (*d*) and length (*SL*) of the occlusive cells, ratio between length and width of the occlusive cells (*SW/SL* ratio), relative water content at stomatal closure (*RWC*$_c$), time elapsed (*t*$_c$) before stomatal closure, moisture content at stomatal closure (*M*$_c$), cuticular transpiration rate (*E*$_c$), and specific leaf area (*SLA*) of the different *Eucalyptus* genotypes under study. Different letters in each column indicate significant differences between clones.

| Clone | $d$ (mm$^{-2}$) | $SL$ (μm) | $SW/SL$ | $RWC_c$ (%) | $t_c$ (min) | $M_c$ (%) | $E_c$ (mmol m$^{-2}$ s$^{-1}$) | $SLA$ (m$^2$ kg$^{-1}$) |
|---|---|---|---|---|---|---|---|---|
| 12€ | 279 ± 7 b | 18.5 ± 0.3 a | 0.77 ± 0.01 ab | 78.2 ± 1.3 ab | 70.2 ± 3.0 b | 61.8 ± 1.0 a | 0.12 ± 0.02 a | 13.4 ± 0.6 c |
| HE | 297 ± 9 b | 23.9 ± 0.3 d | 0.75 ± 0.01 a | 74.8 ± 1.4 a | 80.6 ± 3.7 b | 59.9 ± 0.9 a | 0.27 ± 0.02 b | 9.9 ± 0.3 ab |
| HG | 435 ± 11 c | 21.3 ± 0.3 b | 0.75 ± 0.01 a | 80.9 ± 1.3 ab | 64.5 ± 3.8 ab | 61.7 ± 0.8 a | 0.16 ± 0.02 a | 12.2 ± 0.5 c |
| HI | 431 ± 11 c | 19.1 ± 0.3 a | 0.75 ± 0.01 a | 83.4 ± 1.0 b | 55.5 ± 3.2 a | 61.4 ± 0.8 a | 0.12 ± 0.01 a | 11.0 ± 0.4 bc |
| 225 | 238 ± 6 a | 21.6 ± 0.3 bc | 0.77 ± 0.01 ab | 79.7 ± 1.1 ab | 73.4 ± 3.6 b | 61.0 ± 1.0 a | 0.16 ± 0.02 a | 9.2 ± 0.3 a |
| 227 | 225 ± 6 a | 23.0 ± 0.3 cd | 0.77 ± 0.01 ab | 79.4 ± 1.2 ab | 73.9 ± 4.7 b | 61.4 ± 0.9 a | 0.18 ± 0.03 ab | 9.7 ± 0.4 ab |
| C14 | 226 ± 6 a | 24.0 ± 0.3 d | 0.76 ± 0.01 ab | 79.3 ± 1.0 ab | 76.5 ± 3.6 b | 61.1 ± 0.8 a | 0.18 ± 0.02 ab | 9.0 ± 0.3 a |
| 358 | 204 ± 6 a | 22.2 ± 0.3 bc | 0.77 ± 0.01 ab | 80.4 ± 1.1 ab | 72.3 ± 2.8 b | 60.5 ± 0.8 a | 0.13 ± 0.01 a | 9.3 ± 0.3 a |
| 437 | 235 ± 7 a | 23.2 ± 0.3 cd | 0.78 ± 0.01 b | 80.5 ± 1.2 ab | 72.3 ± 3.5 b | 60.6 ± 0.9 a | 0.18 ± 0.03 ab | 9.1 ± 0.4 a |
| *p* | <0.001 | <0.001 | <0.001 | <0.001 | <0.001 | 0.536 | <0.001 | <0.001 |
| Total | 286 ± 8 | 21.9 ± 0.3 | 0.76 ± 0.01 | 79.6 ± 0.4 | 70.9 ± 1.2 | 61.0 ± 0.3 | 0.17 ± 0.01 | 10.4 ± 0.2 |

**Table 3.** Average values (±SE) of stomatal density (*d*) and length (*SL*) of the occlusive cells, ratio between length and width of the occlusive cells (*SW/SL* ratio), relative water content at stomatal closure (*RWC*$_c$), time elapsed (*t*$_c$) before stomatal closure, moisture content at stomatal closure (*M*$_c$), cuticular transpiration rate (*E*$_c$) and specific leaf area (*SLA*) at each measurement date. Different letters in each column indicate significant differences between dates.

| Measurement Date | $d$ (mm$^{-2}$) | $SL$ (μm) | $SW/SL$ | $RWC_c$ (%) | $t_c$ (min) | $M_c$ (%) | $E_c$ (mmol m$^{-2}$ s$^{-1}$) | $SLA$ (m$^2$ kg$^{-1}$) |
|---|---|---|---|---|---|---|---|---|
| 01 Feb. 2015 | 278 ± 6 bcd | 21.6 ± 0.3 bcd | 0.74 ± 0.01 a | 73.2 ± 1.2 a | 86.3 ± 3.3 c | 58.7 ± 0.6 ab | 0.16 ± 0.01 abc | 11.5 ± 0.3 c |
| 01 May 2015 | 287 ± 7 cd | 22.8 ± 0.3 de | 0.78 ± 0.01 d | 82.9 ± 0.8 c | 81.3 ± 4.3 bc | 60.4 ± 0.6 bc | 0.12 ± 0.01 a | 10.1 ± 0.3 bc |
| 25 July 2015 | 276 ± 10 abcd | 20.7 ± 0.4 bc | 0.78 ± 0.01 bcd | 79.2 ± 1.7 abc | 50.0 ± 2.7 a | 59.4 ± 1.5 abcd | 0.22 ± 0.04 abc | 9.9 ± 0.7 abc |
| 01 Nov. 2015 | 247 ± 8 abc | 18.4 ± 0.4 a | 0.75 ± 0.01 abc | 81.4 ± 2.5 abc | 49.0 ± 1.5 a | 67.7 ± 1.1 f | 0.24 ± 0.03 abc | 14.4 ± 0.6 d |
| 01 Feb. 2016 | 247 ± 7 ab | 22.8 ± 0.3 cde | 0.76 ± 0.01 abcd | 80.8 ± 0.8 c | 66.3 ± 1.6 b | 62.4 ± 0.6 cd | 0.16 ± 0.02 abc | 10.6 ± 0.4 bc |
| 01 May 2016 | 276 ± 10 abcd | 24.2 ± 0.3 e | 0.77 ± 0.01 bcd | 82.6 ± 0.7 c | 64.1 ± 2.7 b | 56.5 ± 0.4 a | 0.14 ± 0.01 abc | 8.1 ± 0.2 a |
| 25 July 2016 | 299 ± 8 de | 26.1 ± 0.3 f | 0.78 ± 0.01cd | 80.5 ± 2.0 abc | 74.7 ± 2.5 bc | 60.8 ± 0.9 bce | 0.18 ± 0.05 abc | 9.2 ± 0.3 ab |
| 01 Nov. 2016 | 229 ± 6 a | 20.3 ± 0.3 ab | 0.75 ± 0.01 abc | 79.4 ± 1.2 bc | 70.2 ± 3.4 bc | 64.6 ± 0.9 def | 0.22 ± 0.02 c | 10.7 ± 0.4 bc |
| 01 Feb. 2017 | 283 ± 7 bcd | 20.2 ± 0.3 ab | 0.75 ± 0.01 ab | 75.5 ± 0.9 ab | 73.9 ± 3.0 bc | 64.5 ± 0.5 df | 0.18 ± 0.02 bc | 11.8 ± 0.4 cd |
| 01 May 2017 | 344 ± 9 e | 21.8 ± 0.3 bcd | 0.77 ± 0.01 bcd | 81.8 ± 0.9 c | 77.7 ± 4.8 bc | 57.3 ± 0.7 ab | 0.12 ± 0.01 ab | 8.3 ± 0.3 a |
| *p* | <0.001 | <0.001 | <0.001 | <0.001 | <0.001 | <0.001 | <0.001 | <0.001 |
| Total | 286 ± 8 | 21.9 ± 0.3 | 0.76 ± 0.01 | 79.6 ± 0.4 | 70.9 ± 1.2 | 61.0 ± 0.3 | 0.17 ± 0.01 | 10.4 ± 0.2 |

### 3.2. Cuticular Transpiration

Concerning the $RWC_c$ and the $t_c$, significant differences were detected among clones (Table 2) and among dates (Table 3). They were also found regarding the interaction between these two factors, with $p < 0.001$ for $RWC_c$ (Figure 3) and $p = 0.001$ for $t_c$. This indicated that the clones' seasonal evolution patterns differed among themselves.

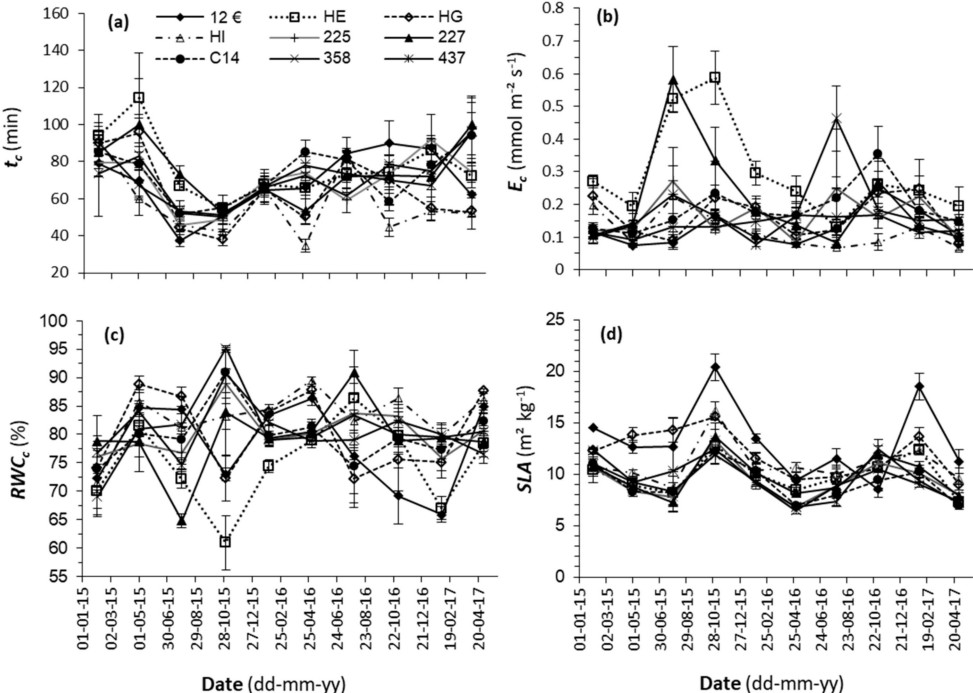

**Figure 3.** Seasonal evolution (average ±SE) of (**a**) the time elapsed until stomata closure ($t_c$), (**b**) cuticular transpiration ($E_c$) based on *LA*, (**c**) relative water content at the time of stomata closure ($RWC_c$), and (**d**) specific leaf area (*SLA*), of the nine *Eucalyptus* clones studied on the 10 measurement dates.

The $E_c$ differed significantly according to clones and measurement dates; both were calculated based on *DW* and *LA* ($p < 0.001$) (Tables 2 and 3). Furthermore, significant differences were found regarding the Clone × Date interaction ($p < 0.001$, Figure 3). The behavior pattern of $E_c$, expressed on a leaf weight basis did not differ significantly from that calculated based on *LA*, in terms of differentiation among clones or among dates, with an overall average value of 1.69 ± 0.06 mmol kg$^{-1}$ s$^{-1}$ of $H_2O$.

Regarding $M_c$, no significant differences were detected among the different clones (Table 2), but they were detected according to dates (Table 3), as well as to Clone × Date interaction ($p < 0.001$). Regarding a general trend, it is worth noting that the lowest values were obtained from the measurements made in May and July ($M_c$ = 56.5%–60.8%) and the highest values were obtained in November ($M_c$ = 64.6%–67.7%).

Finally, *SLA* differed significantly according to the different clones, dates of measurement, and Clone × Date interaction (Figure 3, Tables 2 and 3). Regarding the measurement dates, the clones of *E. globulus* and HE (9.0–9.9 m$^2$ kg$^{-1}$) were different from HI (11.0 m$^2$ kg$^{-1}$) and the latter, in turn, from the group formed by HG and 12€ (13.2–13.4 m$^2$ kg$^{-1}$).

## 4. Discussion

### 4.1. Stomatal Characteristics

The stomata of the *Eucalyptus* clones used in the present study were concentrated on the leaf's abaxial surface. A very small number of stomata were found on the adaxial surface i.e., 2–3 stomata

per square millimeter, in all clones. Tuffi Santos et al. [45], when studying *d* in very young plants of *E. grandis*, *E. urophylla*, *E. saligna*, *Eucalyptus pellita* F. Muell, and *Eucalyptus resinifera* Sm., also found that in the case of the whole species, the adaxial surface (10–80 stomata mm$^{-2}$) had 10 times fewer stomata than the abaxial surface (600 stomata mm$^{-2}$), with differences between taxa. In the present study, the average *d* values of the abaxial face were between 204 and 434 stomata per mm$^2$. This value is within the value range of sclerophyllous leaves, i.e., 100–500 mm$^{-2}$, and is typical in species inhabiting rainforests [46] or temperate zones such as *Pinus taeda* L., *Taxodium distichum* (L.) Rich., or *Ilex cassine* L. [17]. The value is, however, below the 750–1050 mm$^{-2}$ value range found in subtropical species such as *Toona ciliata* M. Roem. [47], or the value of 1000 mm$^{-2}$ reached by some oaks and maples proper to humid temperate zones [17].

Although the studied plants were well watered and fertilized at all times, the leaves that grew mainly in spring, specifically from the end of winter to the beginning of summer (May and July measurement dates) tended to present greater *d*, at least in the case of clones that showed more marked seasonal differences. A higher density would facilitate the water's exit (and the assimilation of $CO_2$) at times of suitable water availability in the soil and non-excessive atmospheric demand [5,35,42]. However, in the case of leaves developed in the middle-end of summer and early autumn (mainly from late July to late September) which correspond to the measurements taken at the beginning of November since leaves from the 3rd–5th whorl were sampled when atmospheric demand was greater *d* decreased, allowing plants to save water and better endure droughts. All this indicates that, apart from the availability of soil water and nutrients, the plants responded to other environmental stimuli (photoperiod, solar radiation, air temperature, relative humidity, etc.) [10,13,48]. The latter controlled the *d* of the new developing leaves at all times, suggesting the existence of an internal mechanism that stimulates and transmits the signal [10]. The clones with the highest *d* studied here, HI and HG, with an average of around 430 mm$^{-2}$, almost doubled the *d* of *E. globulus* clones (358, C14, 437, 227, and 225). The lower *d* of *E. globulus* among the clones studied could be a drought-adaptation characteristic, although other characteristics should be taken into account as a whole [49]. For the *Azadirachta indica* A. Juss and *Populus* species, for example, *d* was positively correlated with net photosynthesis and biomass production for *A. indica* [50] or with gs for *Populus* sp. [51]. However, in other studies, *d* did not significantly affect gs or photosynthetic rates [52]. On the other hand, clones of *E. globulus* and clone HE presented lower plasticity regarding *d* variation throughout the year ($\Delta d < 150$ mm$^{-2}$), while the density in the other three clones varied according to the dates, from 160 mm$^{-2}$ (HG) to 300 mm$^{-2}$ (HI). Therefore, interestingly, these latter clones presented greater plasticity regarding this parameter than *E. globulus* clones.

Generally, in this study, stomatal cell size was smaller in the leaves that grew in the middle-end of summer and early autumn (November measurements), and bigger in the leaves that grew in spring. As in the case of *d*, the reason could be that during the summer and early autumn months, when the temperature, radiation, and photoperiod are higher and relative humidity lower (Table 1), the new developing leaves favored the formation of smaller stomata at warmer and drier times, and vice versa in winter-spring, in order to regulate gas exchange and WUE. The same phenomenon was found in the case of *Sequoia sempervirens (D. Don)* Endl. plants across different plantations in Chile [53]. When comparing the clones, clones containing *Eucalyptus globulus* alleles were significantly larger in size (*SL* = 21.3–24.0 μm, *SW* = 15.9–18.2 μm) than 12€ and HI (*SL* = 18.5–19.1 μm, *SW* = 14.2–14.3 μm). These values were in a slightly higher range than those found for three other *Eucalyptus* species (*E. delegatensis* R. Baker, *E. pauciflora* Sieb. ex Spreng., and *E. radiata* Sieber ex DC.), presenting a range of 9.8–12.0 μm for *OL* and 10.0–14.0 μm for *SW* [49].

On the other hand, considering the size and *d* combination in our study, the clones with the biggest pore opening surface potential, HG, HE, and HI, could have a greater transpiration potential with full open stomata, while at the same time pose a risk of excessive water loss in situations of water shortage when stomatal control was not optimal. It has been reported that *d* and occlusive cell length are related to gs and net photosynthesis, as well as other plant physiological characteristics [11,54–56].

Nevertheless, in other recent studies, no significant correlations between stomatal density, size and the rapidity of response have been detected [13,57]. Consequently, we can assume that the seasonal modifications and adjustments of the stomatal size and $d$ found in this study may affect the plants' physiology, to some extent at least. These modifications surely allow each clone to self-adjust to maintain its best level of photosynthetic efficiency, responding to environmental stimuli. However, the results of this study did not find that $d$ and stomatal size, in themselves, were relevant clone selection criteria for water saving, since in the case of well-watered plants (i.e., $\Psi \geq 1.0$ MPa), maximum transpiration rates were similar between clones (Figures S4 and S5). Therefore, when all pores are supposed to be fully open, apart from the maximum potential of open pore surface determined by $d$ and stomatal size, other factors such as mesophilic conductance, boundary layer resistance, etc., should be taken into account. In addition, water loss, which depends on stomatal opening and other environmental factors, can vary greatly even for plants with a good water status, and this multiple control mechanism seems to have a greater effect on the total amount of water transpired every day than $d$ and stomatal size. For instance, the HE, HG, and 12€ clones maintained high transpiration rates when $\Psi$ dropped from $-1.0$ to $-2.0$ MPa. Meanwhile, the other studied clones began to reduce their transpiration when $\Psi$ reached $-1.0$ MPa (Figures S4 and S5), which indicates a more pronounced water saving behavior. Moreover, in this $\Psi$ ($-1.0$ to $-2.0$ MPa) range, the photosynthetic rate was reduced to a greater extent than E and gs for HE, HG, and 12€ clones, so WUE and intrinsic water use efficiency (IWUE, Figure S6) decreased in the case of these three clones to a greater extent than for *E. globulus* clones in the $\Psi$ range. The latter also points to the worse behavior of the first three clones under moderate water stress conditions.

Studies of different tree species, such as riverside poplars in a semiarid environment [51], rainforest species [46], hardwood species in a subtropical climate [58], and *Eucalyptus globulus* in sites with varying precipitation [59], reported a reduction in stomatal size as $d$ increased. However, in this study, despite finding a significant negative correlation between $d$ and stomata size in the nine clones as a whole ($p = 0.017$), the result was very weak (r = $-0.252$) and not significant for each clone separately ($p > 0.10$).

## 4.2. Cuticular Transpiration and SLA

$RWC_c$ is a notable indicator of the leaves' water state under severe water stress conditions. Water state is closely related to cell turgor and, therefore, it accurately reflects the balance between internal water content, water supply to the leaf and transpiration rate, as well as leaf dehydration tolerance [60,61]. In our study, the seasonal development varied between a minimum of 73.2% (leaves developed in autumn and winter) and a maximum of 82.9% (developed in spring), presenting no significant differences between those developed in spring and in summer. The clones that diverged the most from this general trend and that presented large seasonal oscillations throughout the study were: HE, 227, and 12€. The average values obtained for $RWC_c$, as well as the seasonal development were within the range obtained by Carevic et al. [43], who studied Holm oak trees (*Quercus ilex* L. spp. *ballota*), a Mediterranean species with sclerophyllous leaves. These results are also compatible with the values obtained by other researchers [62,63] for eucalypts, as a range of 79%–90% was obtained, detecting differences between species and between clones. Andivia et al. [64] studied the drought tolerance of two Holm oak provenances and observed seasonal variations for this parameter: they reflected a water conservation strategy during the summer (with $RWC_c$ close to 90%) and water spending during the rainy season. All the above indicates that although the nine *Eucalyptus* clones under study presented slight differences, they all used physiological adaptations to reduce water loss at times of greatest demand, as they reacted by closing the stomata at higher hydration levels ($RWC_c$) at the most unfavorable times from the viewpoint of available water. This latter trait is useful for surviving under climates with dry seasons, such as the Mediterranean climate. Following this line of argument, the series of measurement dates revealed that the HE clone showed the least cautious water conservation behavior: it allowed greater dehydration before stomata closure indicating a survival risk

as water stress progresses, while the reverse was found for the HI clone, and an intermediate behavior was observed for the rest of the clones.

As the leaf is dehydrated, the $t_c$ may vary according to the following factors: the leaf's age; the size of the stoma, and its location; the species; the individuals within a species; the vegetative state; and the environmental conditions under which the measurements are taken [65]. The HI clone stood out among the clones of the present study due to its shorter stomatal closing time (56 min under the measurement conditions), which may indicate a drought resistance strategy but at the cost of reducing growth, differing significantly from seven other clones (225, 227, 358, 437, C14, 12€, HE), whose closing times varied between 70 min (12€) and 80 min (HE). Furthermore, the HG clone showed an intermediate stomatal closing time (65 min). Regarding seasonal development, $t_c$ was included within a range of 49 min (November 2015 measurement) to 86 min (February 2015). In 2015, leaves that formed during periods of greater atmospheric demand took half the time to close their stomata compared to those that formed during colder and wetter periods. This phenomenon was not observed the following year. Further evidence was thus obtained regarding the plasticity strategies employed to acclimatize to conditions such as the Mediterranean climate, and the necessary acclimatization to the environmental conditions of the moment to prevent excessive water loss in the drier months.

Plants' leaf epidermis, especially the cuticle, acts as an effective protective barrier against uncontrolled water loss. During a water stress period, when the stomata are closed, the plants' survival greatly depends on the amount of water lost through the cuticle in question. The $E_c$ values obtained i.e., 0.17 mmol m$^{-2}$ s$^{-1}$ of H$_2$O (1.69 mmol kg$^{-1}$ s$^{-1}$) on average, were higher than those found for full-grown oaks [43] grown in the field but measured also under laboratory conditions (0.06–0.19 mmol kg$^{-1}$ s$^{-1}$). They were, however, below those obtained by Fernández et al. [66] for seven species (*Dichrostachys cinerea* (L.) Wight & Arn., *Populus × euroamericana* (Dode) Guinier "I-214", *Eucalyptus camaldulensis, Casuarina cunninghamiana* Miq.*, Paulownia fortunei* (Seem) Hemsl., *Salix purpurea* L., and *Leucaena diversifolia* (Schltdl.) Benth.) that had grown under shade cloth in a nursery. In this latter study, the values obtained were of 0.83 mmol m$^{-2}$ s$^{-1}$ for *E. camaldulensis* up to 3.98 mmol m$^{-2}$ s$^{-1}$ for *S. purpurea*. The leaves of *Eucalyptus haemastoma* (Sm.), a species belonging to a semiarid summer climate with cold winter nights were analyzed at different temperatures [67], and the $E_c$ values ranged from 0.04 mmol m$^{-2}$ s$^{-1}$ at low temperature (18 °C), to 0.5 mmol m$^{-2}$ s$^{-1}$ at high temperature (38 °C). All these value ranges are within those found in our study. Considering the seasonal variation obtained, the general tendency was that the leaves developed in spring (May measurements) presented the lowest level of $E_c$. Regarding differences between clones, despite showing highly similar $E_c$, one clone (HE), with the highest $E_c$ value for the study period as a whole, differed significantly from five other clones (12€, HG, HI, 225, 358), as it presented an $E_c$ that was 61% above the average of the nine clones. This suggests that when measuring leaf epidermis permeability, the HE clone had a less efficient water saving strategy [64] compared to these five clones in the study and therefore less drought resistance; whereas in the intermediate range, the clones 437, 227, and C14 revealed a moderate water saving strategy. Considering the relationship of $E_c$ with Ψ determined via a water stress test with these clones (Figure S3 for Ψ < −1.7 MPa), most stomata are supposed to be closed since the turgor loss point has been overcome (Table S1). Thus, only $E_c$ would remain. Under these water stress conditions, the HE clones along with HG and 12€, the three clones that may have inherited *E. grandis* alleles, showed the highest $E_c$ rates (Figures S4 and S5), indicating a thinner and/or more permeable leaf epidermis, a parameter that is unsuitable for withstanding periods of water stress. Thus, it is worth studying the relationship between $E_c$ and leaf permeability more in depth, addressing not only leaf and cuticle thickness but also other factors such as the presence of waxes and trichomes which could result in stable selection criteria [68,69]. In addition, concerning the water stress test, only for these three clones (HE, HG, 12€) was the relative water content for which they closed stomata ($RWC_{cs}$) significantly lower than the relative water content in which the loss of cell turgor occurred, $RWC_0$ (Table S1). This would indicate a lower degree of stomatal control, since the stomata do not close even when cell turgor is lost, indicating an unsuitable behavior when the water stress progresses. However, the latter should be

interpreted with caution because $RWC_0$ and $RWC_{cs}$ are measured using different methods and their physiological interpretation differs a little.

Regarding the *SLA*, the *Eucalyptus globulus* clones and the HE clone had the lowest values, differentiating themselves mainly from hybrids 12€ and HG, both with *Eucalyptus grandis* alleles. All this would indicate that *E. globulus* would have thicker leaves, appropriate for its greater adaptation to dry climates, compared to *E. grandis*, typical of more humid climates and drought tolerant [30,70]. Of the two clones that may have inherited both *E. globulus* and *E. grandis* alleles, the *E. globulus* inheritance seems to have dominated regarding this parameter in the case of HE, while in the case of HG, the *E. grandis* inheritance seems to have dominated. The *SLA* values obtained in this study are within the range reported by other authors for eucalypts e.g., 16.1 m$^2$ kg$^{-1}$ (*Eucalyptus occidentalis* Endl.) to 25.4 m$^2$ kg$^{-1}$ (*E. grandis*) [71], and 6.1–8.3 m$^2$ kg$^{-1}$ for plants of *E. dunnii* and *Corymbia citriodora* subsp. *Variegata* (F. Muell.) A.R. Bean & M.W. McDonald aged 11 years [72]. During the seasonal development of the studied clones, *SLA* decreased progressively over time from autumn-winter leaves to spring-summer leaves. This results from the varying climatic conditions of relative humidity, radiation, and temperature, taking into account that the plants were well watered and fertilized, demonstrating once again their sensitivity and ability to react to climatic variables.

## 5. Conclusions

The follow-up, over 2.5 years, of the stomata characteristics and the $E_c$ of nine nursery-grown *Eucalyptus* clones led to the following conclusions:

- All the clones under study showed seasonal variations in *d* and stomatal size, *SLA*, and $E_c$, as well as in $RWC_c$ and the $t_c$, despite the substrate being constantly humidified to field capacity and fertilized, in response to stimuli such as light radiation, photoperiod, temperature, and/or the air's evaporative demand to acclimatize to the environmental conditions.

- Each clone adjusted its own *d* and size values to acclimatize its stomata to the growth conditions. The maximum amount of water transpired with fully open stomata might depend on other internal and external factors. Thus, the criteria of size and *d* alone are not sufficient to differentiate between clones, at least for this study.

- The HE and HG clones presented poor stomatal control as they showed high transpiration rates when Ψ was between −1.0 and −2.0 MPa, which represent a risk in cases of prolonged drought. The HI clone, on the other hand, conserved internal water very efficiently because it closed stomata sooner and at a lesser degree of dehydration.

- Taking into account only the morpho-physiological parameters measured in this study and the genotypes considered, the clone known to resist to droughts, C14, was characterized by low values of *SLA* and $E_c$, high $RWC_c$, low seasonal plasticity regarding *d*, and good stomatal control. These characteristics were shared by the other *E. globulus* clones, though the clones of hybrids differed as they presented less favorable properties for drought resistance such as lower epidermis impermeability (HE, HG, 12€), higher *SLA* (12€, HG), and lower stomatal control under conditions of moderate water stress (HE, HG).

- These analyzed eucalyptus clones showed they had genetic variability for drought resistance. Gains could thus potentially be obtained through the selection and implantation of improved populations. However, in situations of water stress, other morpho-physiological properties should be studied (e.g., WUE, xylem anatomy and cavitation vulnerability, osmotic adjustment capacity, etc.) together with the characteristics studied here.

**Supplementary Materials:** The following are available online at http://www.mdpi.com/1999-4907/11/1/9/s1, Figure S1: (left) Stomata observed on leaf prints taken from the abaxial side of a leaf (400×); (right) cross section of a leaf sampled in the summer of 2017, showing the adaxial side to the right and the abaxial side to the left (400×). Regarding the cross-section of the leaf, four leaves per clone were analyzed in summer 2017 and no significant differences were detected between clones in these four parameters measured: cross-sectional thickness ($p = 0.111$; 267.56 ± 27.4 μm); thickness of the adaxial epidermis ($p = 0.160$; 20.6 ± 2.1 μm); thickness of the abaxial

epidermis ($p$ = 0.370; 17.1 ± 1.6 μm); and palisade parenchyma thickness ($p$ = 0.500; 76.7 ± 7.5 μm), Figure S2: The graph generated by all the pairs of values, *FW*-time (continuous line, rhombuses) and the regression line generated with the points marking a leaf's constant weight drop (dashed line, squares). The cut-off point between the curve and the regression line is assumed to be the $t_c$ (arrow). The slope of the regression line reflects the loss of water over time, from which $E_c$ can be deduced, Figure S3: Plants of the nine clones under study, used to measure daily transpiration (by weighing), and instantaneous transpiration using a portable infrared gas analyzer (Model LCi, ADC, London, UK). Daily transpiration was calculated based on the difference in weight between two measurements taken 24 h apart, measured 1 h after dawn on two consecutive days. The containers were wrapped with white plastic to avoid direct evaporation from the substrate. The total *LA* was measured for each plant. The assay started with plants watered to field capacity, but subsequently, they were watered, every day, with half of the water transpired the previous day, to subject the plants to a slow and progressive process of water stress for 30 days. Ψ was measured exactly at dawn (PMS 1000, Corvallis, USA). The instantaneous transpiration rate (E) was measured 2 h after dawn, when plants show maximum daily transpiration rates. This test was carried out during the summer of 2017, using three plants per clone from the additional plants left over from the main assay, Figure S4: Relationship between daily transpiration rate over a 24-h period, and the water potential at dawn of the first day of each measurement date, for the nine clones studied, Figure S5: Relationship between the E measured 2 h after dawn and the water potential at dawn, for the nine clones studied. E was significantly correlated with gs (E = 9.036 gs + 5.547, r = 0.963, $p$ < 0.001) and net photosynthetic rate, A (A = 0.038 $E^4$ − 0.661 $E^3$ + 3.446 $E^2$ − 2.525 E + 0.619, r = 0.962, $p$ < 0.001). E (mmol m$^{-2}$ s$^{-1}$ of $H_2O$), gs (mol m$^{-2}$ s$^{-1}$ of $H_2O$), A (μmol m$^{-2}$ s$^{-1}$ of $CO_2$), Figure S6. Relationship between the intrinsic water use efficiency (IWUE = A/gs) measured 2 h after dawn and the water potential at dawn, for the nine clones studied. A (μmol m$^{-2}$ s$^{-1}$ of $CO_2$), gs (mol m$^{-2}$ s$^{-1}$ of $H_2O$), Table S1: Mean value (±SE) of the osmotic potential at full turgor (Ψ$_{s100}$) and at the point of turgor loss (Ψ$_{s0}$), the *RWC$_0$* and *RWC$_{cs}$* of the nine studied clones. The measurements were made on two dates, first in well-watered plants and then after the plants were subjected to a progressive water stress test for 30 days in the summer of 2017 (see Figure S3), by means of the construction of isothermal pressure-volume curves, using the methodology described by [73]. $p$: level of significance. Different letters in each column indicate significant differences between clones. *: for each clone, asterisk indicates significant differences between *RWC$_0$* and *RWC$_{cs}$* ($p$ < 0.001, Dunnett's T3 test).

**Author Contributions:** F.R., M.F. and R.T. designed the experiments; A.C. and M.F. conducted the experiment, and wrote the first draft; A.C., J.V.-P. and M.F. analyzed the data; J.V.-P. and R.T. revised and edited the manuscript. All authors have read and agreed to the published version of the manuscript.

**Funding:** This study was supported by the Conselho Nacional de Desenvolvimento Científico e Tecnológico (CNPq)—Brazil (grant number 203224/2014-0), the company ENCE, energía y celulosa S.A., (grant number Contrato art. 68/83) and the National Research Programme, reference CTQ2013-46804-C2-1R and CTQ2017-85251-C2-2-R which, in turn, were financed by FEDER.

**Acknowledgments:** We thank all authors for their contributions to this study. We would like to thank Open Five S.L. services for the English-language revision.

**Conflicts of Interest:** The authors declare no conflict of interest.

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
