# Peer review of "Variability and Plasticity in Cuticular Transpiration and Leaf Permeability Allow Differentiation of Eucalyptus Clones at an Early Age"

_forests, doi:10.3390/f11010009_

Round 1

Reviewer 1 Report

The manuscript of Carignato et al concerns the role of stomatal anatomy and physiology in differences in drought tolerance between fast growing Eucalyptus varieties.  This is an extremely interesting paper given the number of recent studies focused on the role of stomatal dimensions in stomatal responses and as potential mechanisms to improve crop drought resistance.  I am impressed with the dataset concerning drought impacts on stomatal density – this work is quite sparse in the literature.  The manuscript is well written and structured.

For your future work, I hope you can consider using gas exchange systems (eg Ciras-3 or LiCor 6800) to measure stomatal kinetics alongside this valuable anatomical datasets.

I look forward to reading the final version of the manuscript.

Minor suggestions:

Lines 35-39 – stomatal density and size is linked to the degree of control of the stomatal aperture.  This links your two points. See Haworth et al., 2018 Environ. Exp. Bot. 151, 55-63.

Lines 40-43 – regulation of stomatal aperture also varies under drought stress. See Gerardin et al., 2018 Exp. Bot. 153, 188-197 and Haworth et al., 2018 Environ. Exp. Bot. 147, 116-124.

Discussion – perhaps Drake, P.L. et al. J. Exp. Bot. 64, 495-505 would be helpful

Author Response

See attached file: responses to reviewers 1 and 2

Reviewer 2 Report

Dear  Author
I have read the manuscript ‘Forest’ Manuscript ID 64799. Entitle: Variability and plasticity in cuticular transpiration and leaf permeability allow differentiating Eucalyptus clones at an early age. Author did analyzing the selection of genotypes of Eucalyptus based on stomata and cuticular transpiration indicators. Moreover, analysis of these properties with changing the different season with yearly basic with relating the water status, stomatal morphological characteristics such as size and density, SLA and author found the genetic variability, which is much worthy and interesting for the readers and almost meet the standard and match the scope of the journal FOREST.

Overall the manuscript is well written. However, there is many parts where author should give the intension and need to revised carefully with justifiable text. Furthermore, I have some question to ask to the author for more conformation the results because some sentences and result seems not clearly express because- to select the genotypes of Eucalyptus in early age based on cuticular transpiration and stomatal appearance (via. Characteristics, size, density) as well as SLA, are not only enough traits even analysis time even longer (2.5 years). There also should consider other parts should to consider and take into account for genetic improvement for medium drought resistance. Therefore, the manuscripts need to suitably revision and need to modification and should slightly change the story and the presentation style. Some result is interesting to read and those are worthy to reader but only those are not enough at all.

Furthermore, I also cross check your literature citation accordance with the body of your manuscript and found that the some of your references is less match with your title and manuscript script. Those lacking part you should add most relevant reference, some of them I suggested in below comments. “Leaf morphology, anatomy, stem vessel, physiology, and water relation and transpiration” are those highly connected words for your manuscript because those all traits directly related to cuticular transpiration, plant health, productivity, drought resistance and as well as genetic selection and dissemination for long term vision. So, author need to suitable revise of this manuscript with addressing my points, moreover Author should quantify the clones based on their potential characters then only manuscript will worthy to read. In revised version if author revised carefully then manuscript may be acceptable for publication. Now I request to author for Major revision of this manuscript.

My comments and suggesting for the author are following.

Abstract: Abstract is too much poor, I thing you should re-write abstract again based on results of the manuscript text. Please always remember Abstract should always concise, less text and main message of your finding. I personally not much satisfy from your abstract. Even you greatly write the manuscript body but abstract is comparatively poor.

Comment 1.

Abstract Line no. 17-18

“among others, are factors to be taken into account, as well as the clone’s ability to modify them according to growing conditions” == clone ability to modify them? what modify? Moderate drought stress resistance? Yours measured indicators? If so how much extend ? you can not include all but include interested resistance and sensitive clones based on your measured indicators.

Comment 2.

 Abstract Line no. 23-25

“Your result is not clear, what you write in here” please always remember, when you write something it should be under stable for the reader. Moreover, abstract is not like other-text, it is so sensitive part, your main approach is to identify and detail analysis of different Eucalyptus clones but I not see about your result mention with clone, why you are not quantifying the clone for drought resistant based on the transpiration and based on stomata morpholological characteristics (i.e., density and sizes) from your long term study.

 Comment 3.

Abstract Line no. 26

Conclusion: “Significant differences among clones and among measurement dates indicated that there was as much genetic variability” what significant different? You mean all your measurement parameters? (Cuticular transpiration, stomatal size, density increased in all clone/or which clone, and which parameter and what level of significant P value = ?  ). Really very poor writing in here. Please write it clearly. I not make much difficulty but present the pros and constrains of important clones based on your study.

Note: Based on my comments in abstract please modify manuscript suitable and try to modify throught addressing these point.

Comment 4.

 Line 31

Please consider the key words, Remove the “Forest plantation” and add the “Eucalyptus clones”

Introduction

In the introduction there is some noticeable lack of literature citation and related text, somewhat research question and hypothesis as well. that lacking in introduction make the really make of the limiting theme of introduction. Without appropriate literature and questions or hypotheses entirely text make the unclear So please addressed adequately with justifiable literatures citation with adding some new text. Please consider following points in introductions.

Comment 5.

 Line 35

“stomatal morphology (specific leaf area (SLA)” – SLA is not belonging to stomata morphological characteristics; SLA is independent leaf morpho-character, so please improve the text.

Comment 6.

 Line 46-48

“The anatomical variables of stoma size and density appear to be highly sensitive to environmental changes, especially in water stress conditions, possibly due to stomatal resistance to transpiration [8–10]” = please change this way “Stomatal size and density is highly sensitive to environmental abiotic stress such as drought because of stomatal resistance transpiration” in literature citation remove no. 10 literature (Lammertsma, E. I. et al., 2011) because it is not much match in here please remove from here (literature 8 and 9 are okay).

Comment 7.

Line 87-88

You can not say that “morpho-physiological factors more effect than the leaf anatomy for regulation of plant transpiration”. It’s totally wrong, please modify the text like this way with including these two potential literatures. (1) Bhusal et al., 2019 “Impact of drought stress on photosynthesis responses, leaf water potential and stem sap flow of two cultivars…........DOI:10.1016/j.scienta.2018.11.021) and (2) McDowell et al., 2011 “The interdependence of mechanism…. https://doi.org/10.1016/j.tree.2011.06.003) and tresent tem presentation might be different but meaning should like this way “Stomatal conductance is an important physiological factor because close the stomata during drought resulted limit the water loss by transpiration and susceptible of the xylem vessels (i.e. embolism or dysfunction) that resulted in lower hydraulic conductance in plant (Bhusal et al., 2019, McDowell et al., 2011).

Comment 8.

Line 91-92

Before start to write your hypothesis in line no 92. You should include citation related on leaf internal structures and stomatal density for support your study approaches of different clone/species/ cultivars. This way you should improve your text something like this way “Leaf anatomical structures such as palisade parenchyma and stomatal density regulate the leaf functions traits (i.e., photosynthesis and transpiration) which is varied in different cultivars or clone and greater the leaf internal structure greater the net photosynthesis rate and water exchange from the leaf (Bhusal et al., 2018)” You cite this literature. ‘Comparisons of physiological and anatomical characteristics between two cultivars of bi-leader apple trees’ DOI:10.1016/j.scienta.2017.12.006. Please also consider the flow of this paragraph with above and below eg. last paragraph (81 to 99 line).

Comment 9.

Line 131-133

The sentence is much wordy, please change the structure of the sentence, you should write simple way ‘what characters you consider before taking the leaf sample’

Comment 10.

Line 137-138

The sample leaves were considered to have developed during the 90-day period prior to each measurement date and in table 1 (one) you showed climatic variables in the study site but please also consider the reason for taking those parameter, what contribution of those climatic data on leaf and its stomatal characteristics and why you consider those data, please also include 1-2 reasonable sentence in that paragraph.

Comment 11.

Line 232.

Please consider the full form of the title what you use abbreviation in the table, you cannot write the abbreviation in title and inside the table, please write full with abbreviation in the title of the table and then only use abbreviation inside the table. Please always remember that each table are independent, in scientific writing it’s not considering.

 Comment 12.

Line 264-268

You present the result of d (stomatal density) of different clone but I not saw the methodology while you taking this count. How many field in one leaf, how many leaf sample you took this account? this all information about method and procedure (even brand of microscope, magnification, sample replication and so on) you should include in material and methods.

Comment 13.

Line 273-282

You write many sentences without just citation from the line of 276-282, please give the citation, to support of your text. You may include the citation in line 276-277 “higher the stomatal density higher the CO2 assimilation rate………” and cite the literature (ie. Bhusal et al., 2018) or any other potential referenece related on comparing cultivar or species, or clones.

Comment 14.

Line 295-297

Stomatal cell size was smaller in the leaves that grew in summer (November measurements), November is summer? Please check it. And also give this reason why smaller in summer and why larger in spring, write logically if available give the citation. You wrote here due to “Acclimatization” it’s true but only this is not enough.

Comment 15.

Line 357-359

Similar to like this 3 line please try to quantify the clone of medium drought resistance throughout the manuscript. For eg. These 3 line you written well where you quantify the drough and transpiration point of view, I will aspect in newly revised version you will quantify clone and possibly modify the whole manuscript, I not make you much trouble but grading and quantification, which good? Which poor? Which medium? its really need based on your nature of the experiment, then only it will valuable of your work, as well interesting for the readers and then only it will helpful for the genetic improvement program for commercial cultivation Eucalyptus.

Comment 16.

Line 419

Please reduce conclusion, it’s too long, just focus the positive and negative aspect of potential clones, its too detail, it is not mean that you did long season experiment and include all result in conclusion section, give main point or conclusive remarks of clones in including main and interesting things from your finding.

Comment 17.

Line 506

Reference: please double check the citation, format, language error everything in freference as well as whole manuscript smoothly. While you revising manuscript please delete the less valuable old citations and if you fell this will good for cite then you are welcome to cite few of new references but you should mark which literature you delete and which literature you add please notify in new revision. Good Luck and thank you!

Author Response

(The authors gave the same response as above.)

Round 2

Reviewer 2 Report

Dear Author, 

I thoroughly reviewed your revised manuscript, now the manuscript is significantly improved. You addressed all the previous quarries and suggestions for the significantly improve the manuscript. Apart from this, you address another reviewer and editor comment as well. All of these I checked. Now your manuscript has good of story flow and new thing and it is interesting to the reader.

Now I am going to suggest to the editor for accept this manuscript for publication in the FOREST journal. Incoming new research if you compared the Eucalyptus clone along with seasonal comparison of photosynthesis and study of stem anatomical features (vessel structures) along with cuticular transpiration this will be another interesting chapter for you. Many thank you.